# Supporting the Community’s Health Advocates: Initial Insights into the Implementation of a Dual-Purpose Educational and Supportive Group for Community Health Workers

**DOI:** 10.3390/healthcare13243288

**Published:** 2025-12-15

**Authors:** Marcie Johnson, Kimberly Hailey-Fair, Elisabeth Vanderpool, Victoria DeJaco, Rebecca Chen, Christopher Goersch, Ursula E. Gately, Amanda Toohey, Panagis Galiatsatos

**Affiliations:** 1Medicine for the Greater Good, Johns Hopkins School of Medicine, Baltimore, MD 21224, USApgaliat1@jhmi.edu (P.G.); 2Johns Hopkins Bayview Medical Center, Baltimore, MD 21224, USA; 3Krieger School of Arts and Sciences, Johns Hopkins University, Baltimore, MD 21218, USA; 4Division of Pulmonary and Critical Care Medicine, Johns Hopkins School of Medicine, Baltimore, MD 21224, USA

**Keywords:** community health workers, health workforce, professional burnout, peer group education, program evaluation

## Abstract

**Background/Objectives:** Community health workers (CHWs) play a critical role in advancing health equity by bridging gaps in care for underserved populations. However, limited institutional support, inconsistent training, and lack of integration contribute to high rates of burnout. The Lunch and Learn program was launched in Maryland in fall 2023 as a virtual continuing education and peer-support initiative designed to foster professional development, enhance connections among CHWs, and align with Maryland state CHW certification requirements. This article describes the program’s first year of implementation as a proof-of-concept and model for scalable CHW workforce support. **Methods:** The program offered twice-monthly, one-hour virtual sessions that included expert-led presentations, Q&A discussions, and dedicated peer-support time. Participant engagement was assessed using attendance metrics, post-session surveys, and annual feedback forms to identify trends in participation, learning outcomes, and evolving professional priorities. **Results:** Participation increased over time with the program’s listserv expanding from 29 to 118 members and average session attendance more than doubling. CHWs highlighted the program’s value in meeting both educational and emotional support needs. **Conclusions:** The Lunch and Learn program demonstrates a promising model for addressing burnout through education and community connection. As an adaptable, CHW-informed initiative, it supports both professional growth and well-being. Ongoing development will focus on expanding access, incorporating experiential learning assessments, and advocating for sustainable funding to ensure long-term program impact and CHW workforce stability.

## 1. Introduction

Community health workers (CHWs) are frontline public health professionals who bridge critical gaps in care for underserved populations and play an indispensable role in advancing health equity. CHW programs have long served as a cornerstone of community-based care globally. For example, in Latin America, CHWs were central to the *promotores de salud* model, which emphasized peer-led health promotion and social empowerment within marginalized communities [1]. Similarly, the United Kingdom’s National Health Service (NHS) has expanded outreach models that integrate CHWs into multidisciplinary teams to strengthen primary care access and population health management [2].

The American Public Health Association defines community health workers (CHWs) as frontline public health professionals who either belong to the community they serve or have a strong understanding of it [3]. CHWs’ trusted relationships with the community allow them to connect individuals with health and social services, thereby improving access, service quality, and cultural relevance. In addition, CHWs promote individual and community health by providing outreach, education, informal counseling, social support, and advocacy to build knowledge and self-sufficiency [4]. CHWs play an integral role in community health and healthcare, complementing an overburdened health workforce, increasing availability of and access to basic health services, and organizing communities to address social determinants of health, especially in underserved populations [4,5,6].

The definition of a CHW is necessarily broad, allowing CHWs to address the diverse needs of the communities they support. Their roles can vary widely, encompassing work in primary care settings, health education, and community-based organizations and is often driven by the specific needs of the client populations or employer funding sources [7]. However, this flexibility in definition can present challenges to the workforce, preventing full integration into healthcare settings and suboptimal utilization [8]. Without a clearly defined role within a clinical team, CHWs are often underutilized or face uncertainty around their responsibilities [7]. This lack of clarity may contribute to feelings of overwhelm or hinder effective collaboration with their clinical peers, which are known contributors to high rates of burnout in the healthcare workforce [9].

Burnout is a type of work-related stress characterized as emotional exhaustion resulting in depersonalization and decreased feelings of personal accomplishment in a professional context [10]. There are various significant, and potentially long-term consequences of burnout [11]. For example, research shows a correlation between professional burnout and symptoms of anxiety and depression, driven by emotional exhaustion [12]. Healthcare workers, including CHWs, may be particularly susceptible to “compassion fatigue” and burnout since they frequently interact with patients who have complex physical and social needs [11,13].

It is crucial to address burnout and construct a healthy work environment to ensure high-quality patient care to develop a resilient healthcare system [14]. Multi-pronged approaches involving individual and organizational-level strategies have been found to be most effective in mitigating burnout [9,15]. In addition to adequate staffing and manageable workloads, protecting healthcare workers’ physical and mental health through individual-focused interventions like mindfulness, stress management, and small group discussions can be helpful in reducing burnout [9].

Another way to address burnout is through professional development opportunities. Healthcare employers can reduce CHW burnout through consistent access to professional development opportunities, training, and educational resources to help workers feel greater confidence, engagement, and motivation [9]. Research shows that increased autonomy and control over the work environment can be protective against burnout. However, emphasizing personal resiliency without addressing organizational resiliency may foster a culture of self-criticism and alienation rather than encouraging system-level support programs and resources [9,16]. Building resilient healthcare providers necessitates a multifaceted and sustained approach to developing and implementing effective interventions that treat burnout mitigation as a shared responsibility between healthcare systems and individual providers [16].

The current work presents a proof-of-concept description of an innovative program designed to support CHWs through education and peer connection. The Lunch and Learn program was created in direct response to CHWs’ expressed needs for professional support, addressing persistent gaps created by incomplete integration and lack of standardization within the CHW workforce. This paper provides a descriptive account of how the program was conceptualized, built, and implemented over its first year of operation. Rather than serving as a formal evaluation, this work highlights lessons learned, program strengths, and areas for growth, offering an adaptable model-in-practice for organizations seeking to strengthen CHW engagement, professional development, and well-being.

## 2. Methods

This project was designed as a descriptive, practice-based implementation report intended to document the creation and feasibility of the Lunch and Learn model. It was not structured as a formal evaluation or hypothesis-testing study.

### 2.1. Program Description

The Lunch and Learn Program was introduced in the fall of 2023 to CHWs in the state of Maryland as an innovative approach to providing CHWs with continuing education while fostering a sense of professional community and support. To become a certified CHW in Maryland, a person must complete an accredited CHW training program that includes 100 h of didactic work and 40 h of practicum. This training must cover nine core competencies for CHWs, outlined by the Maryland Department of Health, including: 1. Advocacy and community capacity building skills; 2. Effective oral and written communication skills; 3. Cultural competency; 4. Understanding of ethics and confidentiality issues; 5. Knowledge of local resources and system navigation; 6. Care coordination support skills; 7. Teaching skills to promote healthy behavior change; 8. Outreach methods and strategies; and 9. Understanding of public health concepts and health literacy. To renew certification, CHWs must complete 20 h of continuing education credits every two years that address these competencies and foster professional development.

The aim of the Lunch and Learn Program was two-pronged: 1. provide continuing education to CHWs; and 2. foster a professional community for CHW support to help mitigate burnout. To accommodate these aims, Lunch and Learns were held virtually twice a month for an hour. Each session was structured to include a 30 min presentation by an expert speaker or a representative from a community-based organization, followed by a Q&A session allowing CHWs to engage with the speaker and clarify key takeaways; and a 10 min open discussion (“holding space”) for CHWs to share experiences, discuss ongoing challenges, and seek peer support.

The first 50 min of each session (i.e., the presentation and Q&A) were recorded, with participant permission, so that they could be viewed by those who could not join live. A dedicated CHW Professional Resources repository was also created to house session recordings, presenter slides (when available), and Appendix A. Following each session, an updated link was distributed via a CHW listserv, which is an email distribution list, allowing CHWs to build a “rolodex” of professional resources for reference and ongoing learning.

Presentation topics were selected using input from CHWs on areas of interest (Table 1).

### 2.2. Developmental Evaluation Approach

This project employed a developmental evaluation framework to assess feasibility and inform real-time adjustments during the first year of implementation. Data from surveys and post-session polls were used to monitor participation patterns and collect formative insights rather than to conduct formal hypothesis-driven analysis.

Surveys were conducted in January 2024 and January 2025 to gather CHW input regarding building, implementing, and modifying the Lunch and Learn Program. Additional questions regarding the format of the Lunch and Learns and feedback about the program were obtained in the 2025 survey with the intent of shaping and expanding the program based on the input of participants. Because this project was a formative, practice-based implementation, the survey instruments were not psychometrically validated. The survey questions were created collaboratively by program facilitators based on CHW core competencies established by the Maryland Department of Health, prior program experience, and participant feedback from earlier sessions. No formal pretest was conducted because the surveys were designed for formative feedback rather than research validation. Surveys were sent to all CHWs on the Lunch and Learn listserv.

The surveys included both ranked-choice and open-ended questions to capture insights about participants’ priorities and experiences. For the ranked-choice items, participants were asked to rank available options (e.g., goals, topic areas) in order of importance. Rankings were scored automatically using Qualtrics software (Version January 2025), which assigns a numerical value to each position and aggregates results across respondents to determine the average ranking for each option. Lower mean values indicate more importance. Open-ended questions were reviewed manually to identify broad themes and notable insights, following a simplified thematic review approach without formal coding software. Responses were used to highlight patterns, recurring ideas, and unique insights relevant to program development and refinement.

Beginning in May 2024, CHW Lunch and Learn participants were asked to complete a brief poll at the end of each session. This poll asked CHWs to indicate which of the nine core competencies for CHWs, outlined by the Maryland Department of Health, were addressed in the session that day and to list one new piece of information learned that day. Please see Appendix A for complete survey instruments and post-session polls.

### 2.3. Engagement Data

Listserv growth since November 2023 was examined. Zoom, Version 6.6.10, data were exported and descriptive analyses were conducted using Microsoft Excel, Version 16.013.4, software to determine live attendance patterns and engagement during Lunch and Learn sessions. Exported zoom attendance logs provided information that was used to determine how many sessions each person attended and for how long they attended, allowing for tracking of both individual and aggregate participation over time. Individuals who joined a session for less than 15 min (approximately 25% of the one-hour session) were excluded from analyses to ensure that only meaningful attendance was captured. The first 15 min of each session is generally dedicated to participants joining the call and announcements and are therefore not representative of meaningful content. Presentations typically begin around 15 min into the call.

For each session, we calculated the total number of attendees and the average duration of attendance. We also examined the trailing average session attendance, which represents the rolling mean number of participants across successive sessions—an approach that smooths short-term fluctuations and more accurately illustrates trends in attendance over time.

In addition, we calculated the percentage of sessions attended by each participant relative to the total number of sessions offered. This was used to estimate the average proportion of sessions joined after an individual’s first attendance, providing a measure of repeated participation and sustained engagement with the program.

### 2.4. Ethics and Institutional Review Board (IRB) Statement

The Johns Hopkins University School of Medicine Institutional Review Board has approved this study as “exempt” on 1 November 2022 (IRB0035506) in accordance with 45 CFR 46.104(d)(1) which states that “Research conducted in established or commonly accepted educational settings, involving normal education practices that are not likely to adversely impact students’ ability to learn required educational content or the assessment of educators who provide instruction” can be considered exempt. Informed consent for participation was obtained from all subjects involved in the study.

## 3. Results

### 3.1. Survey Completion

In January 2024, the survey was distributed to 52 individuals on the Lunch and Learn listserv, and in January 2025, it was shared with the same listserv, which had grown to 117 members (118 after a reshare two weeks later). In 2024, 18 people (35%) completed the survey and in 2025, 23 people (19%) completed the survey.

### 3.2. Participant Goals for Lunch and Learn Program

Participants ranked the importance of various goals in 2024 and 2025 (Table 2). In 2024, participants ranked networking as their top priority for the Lunch and Learn, followed by getting support from fellow CHWs, problem solving/case work sharing, and earning continuing education credits. In 2025, participants ranked problem solving/case work sharing as most important, followed by getting support from fellow CHWs, earning continuing education credits, and networking.

In addition, in both 2024 and 2025, using the Lunch and Learns for learning about resources emerged as important. Participants were given the opportunity to provide an open-ended response, elaborating on other goals that mattered to them in the context of the Lunch and Learn program. Responses captured goals including continued learning about health topics and resources (education), collaboration (networking), and career development (professional development). One respondent eloquently captured these goals in their statement:


*“I aim to establish best practices and share knowledge to enhance the support we provide to our clients. Additionally, I am seeking professional growth opportunities, as there are limited avenues for career advancement as a Community Health Worker (CHW) in Maryland. I hope to find opportunities that will contribute to both my professional development and the success of the organization.”*


### 3.3. Topics of Interest for Lunch and Learn Program

Participants also ranked areas they wanted the Lunch and Learn series to focus on with more detailed information (Table 3). In 2024, respondents indicated that learning more about medical system information was most important, followed by getting support for CHWs, learning about community resources, learning about health topics, and professional development. In 2025, respondents ranked learning about community resources as most important, followed by learning about health topics, professional development, learning about medical system information, and getting support for CHWs.

To foster engagement in the Lunch and Learn program, we asked participants to suggest topics of interest for the presentations (Table 4). Topic ideas spanned various health areas, requests to learn more about insurance, requests to hear from community organizations and learn about resources, and requests for professional development and CHW support.

### 3.4. Additional Supports for Lunch and Learn Program

When asked how the Lunch and Learn program could be used to best support CHWs, respondents noted logistical support such as sending reminder emails about the program, maintaining the listserv, and providing access to recordings of the meetings. Responses also captured requests for both virtual and in-person learning opportunities, development of a central resource repository, additional opportunities to share and exchange resources, and continuing to bring in subject experts for presentations.

### 3.5. Feedback and Revisions of the Lunch and Learn Program

In the 2025 survey, respondents were asked to provide feedback regarding the Lunch and Learn program that will be used to revise the program moving forward. Participant feedback regarding the Lunch and Learn series in 2024 was overwhelmingly positive (see small sample of examples below):


*“I just appreciate that there is a place for CHWs to gather.”*



*“I love this team and thank you so much.”*



*“You are the best support I have, so thank you.”*


Respondents noted feeling that they learned about new resources, gained education and knowledge, and felt the program provided helpful information on a wide variety of topics. In addition, respondents stated that they liked having expert-led sessions. They also indicated that they enjoyed sharing information and interacting with other CHWs.

When asked about making changes for the 2025 Lunch and Learn program, respondents proposed incorporating breakout sessions into the format of the program and offering evening sessions for those who cannot attend at lunch. In total, 91.7% of respondents indicated that they supported changing the Lunch and Learn program format so that one meeting a month is dedicated to CHW case check-ins, resource exchanges, and support.

### 3.6. Evaluation of Core Competencies and Learning

To evaluate how Lunch and Learn sessions supported CHWs’ professional development in Maryland’s nine core competencies for CHWs, attendees identified the addressed competency after each session. Overall, attendees felt that Lunch and Learn sessions addressed competencies in the following order of frequency (with the first two being tied): 1. (Tie) Knowledge of local resources and system navigation; 2. (Tie) Advocacy and community capacity building skills; 3. Teaching skills to promote healthy behavior change; 4. Understanding of public health concepts and health literacy; 5. Care coordination support skills; 6. Cultural competency; 7. Outreach methods and strategies; 8. Effective oral and written communication skills; and 9. Understanding of ethics and confidentiality issues.

Learning was evaluated beginning in May 2024 using long form responses. Examples of information learned can be found in Table 5.

### 3.7. Lunch and Learn Engagement Data

The Lunch and Learn program listserv grew from 29 people in November 2023 to 118 people as of January 2025, quadrupling in size.

Zoom data were analyzed to determine how many people joined each Lunch and Learn session live and how long participants stayed on the call. On average, 18 people attended each Lunch and Learn session live. Participants remained on the call for an average of 48.7 min. As illustrated in Figure 1, session attendance grew over the first year of program implementation from an average of 11 attendees in the first month of the program to an average of 24 attendees in the most recent month of the program.

In addition, zoom data showed that participants regularly came to more than one Lunch and Learn session. Figure 2 shows that most individuals returned to 20–40% of Lunch and Learn sessions following the initial session they attended.

## 4. Discussion

### 4.1. Program Impact and Findings

Community health workers are indispensable to the public health ecosystem, particularly in underserved communities where they help bridge critical gaps in care. However, without adequate institutional support and meaningful integration into healthcare systems, CHWs are left vulnerable to high levels of burnout. This burnout not only impacts their well-being but also undermines the sustainability and effectiveness of the services they provide. Research shows that CHWs with higher levels of anxiety and lower levels of job satisfaction are more likely to report higher levels of burnout and compassion fatigue [13]. The Lunch and Learn program illustrated in the current work demonstrates a promising approach to addressing some of these challenges by fostering professional development, peer connection, and a sense of community among CHWs, which are known protective factors against burnout.

The current work describes how our Lunch and Learn program was built and implemented over its first year. The program was launched in fall 2023 to support Maryland CHWs by offering continuing education and fostering a professional community to help combat burnout. Designed to align with state certification requirements, the program provided virtual sessions twice a month featuring a 30 min expert presentation on CHW core competencies, followed by a Q&A and a 10 min open discussion for peer support. Topics were chosen based on CHW input to ensure relevance and engagement, making the program both responsive and practical in addressing workforce needs.

### 4.2. Participant Engagement and Professional Development

Research shows that community participation and stakeholder engagement is critical for the sustainability of community-based programs, such as the Lunch and Learn series [17,18]. To continually tailor and improve the Lunch and Learn program, participant feedback was collected through annual surveys in January 2024 and 2025, as well as brief post-session polls starting in May 2024. These feedback tools captured insights from CHWs about their goals, topic preferences, and support needs, and were instrumental in shaping the program’s content and format. The 2025 survey also included questions specifically focused on the structure and impact of the sessions. Additionally, CHWs were regularly asked to reflect on which core competencies were addressed and to identify new information gained, ensuring that each session remained aligned with certification requirements and participant interests. This approach allowed for active participation by CHWs to help refine and curate the Lunch and Learns, fostering trust between the CHWs and the program facilitators and cultivating a sense of ownership over the program.

To enhance engagement and ensure relevance, the Lunch and Learn program actively solicited topic suggestions from CHWs, resulting in a diverse array of requests across health education, insurance navigation, community resources, and professional development topics. Health topics remained a major focus, with interest in areas such as mental health, autism, chronic disease management, and LGBTQ+ health. Participants also expressed a strong desire to learn more about navigating insurance systems, including Medicare, Medicaid, and resources for uninsured patients. Requests for presentations from community-based organizations highlighted a need for practical, localized information—such as housing resources, transportation support, and services for vulnerable populations. Looking at zoom attendance data, we saw increased rates of participation across the first year of the program, with average session attendance doubling over time. In addition, data also revealed that participants consistently returned to multiple Lunch and Learn sessions, illustrating investment and engagement in the program.

Survey results from 2024 and 2025 revealed evolving priorities among Lunch and Learn participants. In 2024, CHWs ranked networking as their top goal, whereas in 2025, the emphasis shifted toward problem solving and case work sharing. Despite the shift in rankings, support from fellow CHWs remained a consistently high priority across both years. Through mechanisms such as interacting with others in similar situations, sharing knowledge, fostering sense of belonging, increasing self-confidence, providing a support structure, providing symptom relief, and fostering behavioral change, peer support has been shown to alleviate work-related stress and burnout in healthcare professionals [19].

Interestingly, goals that emphasized connection and collaborative learning were generally ranked higher than receiving continuing education credits. Open-ended responses reinforced themes of education, collaboration, and career advancement. Areas of additional interest also shifted over time. In 2024, participants prioritized learning about the medical system, whereas in 2025, the focus shifted to community resources and health topics. These insights underscore the Lunch and Learn program’s value not just as a means of fulfilling certification requirements, but as a dynamic space for professional connection, resource sharing, and personal development.

Survey responses also highlighted an emphasis on professional development needs, including resource navigation, policy advocacy, and creating stronger statewide networks for collaboration. These suggestions underscored the value of a responsive, CHW-informed curriculum that supports both client care and workforce sustainability. CHWs often face significant frustrations and concerns, including lack of professional development opportunities, being excluded from systems-level decision-making due to a lack of acknowledgement for their role, poor or no compensation for their work, and lack of reimbursement for out-of-pocket costs [20]. Creating programs and initiatives, such as the Lunch and Learn program, may be a critical part of addressing some of the unmet needs of this workforce. In addition, training other healthcare workers to more effectively engage and collaborate with CHWs will be important in integrating CHWs into the workforce [21]. Expanding public understanding of how CHWs can best be utilized can include community education and outreach on the role of CHWs and their capacity to complement health services [21].

To assess how the Lunch and Learn sessions contributed to CHWs’ professional development in Maryland’s core competencies, attendees identified which competencies were addressed after each session. Long-form responses were used to evaluate learning outcomes from the Lunch and Learn sessions, with participants noting valuable new information gained across topics such as autism, Alzheimer’s disease, breast cancer, and resources like MyChart, as well as professional development strategies to manage burnout. CHWs reported that the sessions most frequently addressed competencies related to “knowledge of local resources and system navigation” and “advocacy and community capacity building skills”, which tied for the top spots. Other competencies frequently covered included teaching skills for promoting healthy behavior change, understanding public health concepts and health literacy, and care coordination support skills. Less frequently addressed competencies included cultural competency, outreach methods, effective communication, and ethics and confidentiality issues. This evaluation suggests that the program effectively aligned with key areas of CHW professional development, though certain competencies may benefit from further focus in future sessions.

We also garnered feedback to help understand the strengths and weaknesses of the program. Participants indicated that programmatic strengths included expert-led presentations and an environment that fosters collaboration and mutual support among CHWs. Logistic strengths included sending reminder emails, maintaining the listserv, and providing access to session recordings. Possible areas for growth included requests for more flexible learning formats (both virtual and in-person), the development of a central repository for easier access to community-based resources, incorporating breakout sessions into the meetings, and offering evening sessions for greater accessibility. Additional ways to cultivate community include building opportunities for CHWs to learn more about each other, encouraging communication beyond biweekly meetings, and creating circumstances for voluntary interaction beyond class requirements [22]. These efforts could be supported by modeling and participation from veteran CHW attendees of the Lunch and Learn program [22]. A study examining community building and relationships in the context of virtual learning cited missed opportunities for informal relationship building that may encourage community building and suggested finding opportunities to bring people together in-person [23]. Future efforts to gather participant location data on Zoom could provide a better understanding of geographical dispersion of CHWs across Maryland to facilitate accessible in-person programming and events.

### 4.3. Global Relevance and Theoretical Framework

Viewed in a broader context, the experiences of CHWs in the Lunch and Learn program can be interpreted through global frameworks of knowledge sharing and professional networks, including concepts such as mindlines and communities of practice [24,25,26]. The Lunch and Learn is an example of how structured CHW programs and peer networks can strengthen professional capacity. CHWs, like many healthcare professionals, often rely on internalized knowledge or collectively reinforced understandings of care, a concept known as mindlines [24,25]. Through shared experiences, practical strategies, and peer feedback, CHWs build a mental toolkit of effective approaches to address community needs. By participating in regular peer interactions, they also form communities of practice, which support knowledge exchange, professional growth, and improve client outcomes [26]. Recognizing these frameworks provides context for initiatives like the Lunch and Learn program, which foster both individual and collective expertise.

International health programs that integrate and prioritize CHWs, for example, in Latin America and the United Kingdom [1,2], also highlight the value of structured CHW programs and peer networks. Our findings similarly underscore that supporting CHWs through professional development and collaborative learning may be essential for sustaining community-based care. Research shows that equipping CHWs with comprehensive knowledge of local resources allows them to feel more confident in their abilities to address the diverse needs of their communities. This preparedness enables them to offer timely and culturally appropriate support, which is fundamental to building trust and ensuring effective service delivery. For example, CHWs in Latinx communities have described themselves as “puentes” or bridges, emphasizing their role in connecting community members to essential health and social services resources through shared experiences and cultural understanding [20]. Literature on training CHWs in low- and middle-income countries further suggests that context-specific training translates to greater quality of care [21].

In the context of mindlines, communities of practice, and ongoing international CHW efforts, the relevance of the Lunch and Learn program is apparent. This program is a valuable proof-of-concept for fostering CHW professional growth, peer learning, and community-informed practice. By providing a structured forum for knowledge exchange and collaboration, the program demonstrates how CHWs can build both individual capacity and collective expertise, working toward strengthening community-based care at local and broader levels.

### 4.4. Limitations

While the current program highlights the workability of responsive, community-informed strategies that honor the unique needs of CHWs, there are various important limitations. Primarily, it is important to note that the surveys and polls were brief and intended to capture CHWs’ initial thoughts and feedback about this type of professional gathering. As such, we did not collect comprehensive demographic data or retention metrics and cannot draw related conclusions. In addition, the current sample size of respondents was relatively small. Future research should aim increase sample size and explore additional variables, such as demographics, retention rates, care coordination impact, and client-level impacts. Another important limitation to note is that all patterns and insights drawn from survey responses were analyzed manually. Given the limited depth of qualitative data, no formal coding software, multiple coders, or reliability measures were used.

### 4.5. Future Directions/Implications

As the program develops, future aims include more systematic evaluation of the Lunch and Learn program. The current work aimed to serve as a proof-of-concept and model for building a responsive, CHW driven program. In the future, systematically collecting additional data, including possible secondary workforce data that can be used to link Lunch and Learn participation with retention and well-being metrics, will be helpful and important. In addition, more systematically investigating the effectiveness of the Lunch and Learn program, in terms of learning and meeting core competencies, will be important. Trends in healthcare encourage using hands-on performance-based assessments rather than self-report measures to demonstrate competence [22]. Recent CHW program evaluations have employed a combination of competency-based and experiential learning approaches to assess CHW learning [27]. Over the years, CHW training has evolved to place greater emphasis on application of knowledge through simulations, supervised practice, demonstration of skills under observation, and peer assessments [21]. Future evaluation tools should be developed to assess the impact of the Lunch and Learn program using more experiential and practice-based measurements.

Going forward, it will be essential to continue evaluating the impact of interventions like the Lunch and Learn while advocating for broader systemic changes that promote both individual resilience and organizational responsibility for CHWs. Sustained investment in the CHW workforce is not only a matter of occupational health, but a crucial component of equitable and effective public health delivery. Currently, funding for CHW services is primarily short-term and inconsistent, including sources like grant funding, nonprofit contributions, general revenue, and reimbursement for services [28]. Temporary and unstable funding sources are unsustainable and disadvantageous to CHWs and the communities they serve because they affect CHWs’ abilities to work continuously within their communities without interruption in care [8,28]. Discontinuing or abandoning programs may result in lowered levels of community support and trust in research and public health or medical institutions. This subsequently undermines the function of CHWs in building trust between underserved communities and the healthcare system [28]. Stable funding would give organizations confidence that they can integrate CHWs into care teams without workflow disruptions caused by funding fluctuations [8]. The positive impact of long-term investment in CHWs is demonstrated by improved health outcomes and better CHW retention in states that have invested into CHW programs [29]. It is imperative to continue to communicate to stakeholders that sustained investment in CHWs is simultaneously financially beneficial and significant for improving patient outcomes and achieving health benefits.

## 5. Conclusions

In conclusion, the Lunch and Learn program represents a promising model for supporting CHWs through education and peer engagement. By continuing to refine the program based on participant feedback, integrating innovative learning strategies, and advocating for systemic workforce support, similar initiatives can play a crucial role in enhancing CHW effectiveness and well-being. Future research should focus on scalability, long-term workforce outcomes, and integration with broader healthcare training initiatives to maximize the program’s impact.

## Figures and Tables

**Figure 1 healthcare-13-03288-f001:**
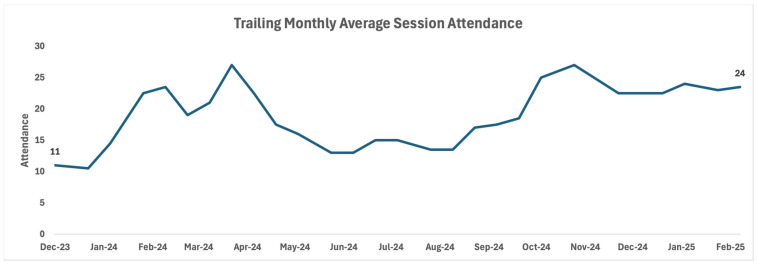
Trailing monthly average session attendance over time.

**Figure 2 healthcare-13-03288-f002:**
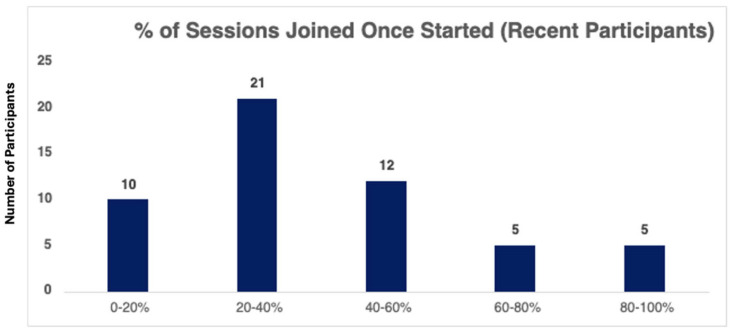
Percent of repeat sessions attended.

**Table 1 healthcare-13-03288-t001:** Topics presented during Lunch and Learn Program from November 2023–February 2025.

Themes	Topics
Health and Disease Topics	Narcan Training Overview Nutrition and Children’s Health Addressing Asthma & Lead Triggers in the Home Breast Cancer Screening Resources Lead Pipes and Water Cancer Screenings Vaccines Overdose/Opioid Crisis: How We Got Here and What Can Be Done Immunodeficiency Disorders
System and Resource Navigation	The Culture of Hospitals Outpatient Care Community Pharmacy Resources Hospital Resources and Survey Enrolling in Medicare/Enrolling in Private Insurance SNAP Benefits Advanced Care Planning Helping Clients Navigate MyChart
CHW Support and Professional Development	Health Literacy CHWs and Research CHW Burnout CHW Case Presentations CHW Roles Supporting Individuals with Rare Diseases
Community and Organizational Resources	Autism Society of Maryland/Pathfinders Caregiver Support Alzheimer’s Association of Maryland

**Table 2 healthcare-13-03288-t002:** Goals ranked in order of importance.

	2024	2025
Most important	Networking	Problem solving/case work sharing
	Support from fellow CHWs	Support from fellow CHWs
	Problem solving/case work sharing	Earning continuing education credits
Least important	Earning continuing education credits	Networking

Note: Presented order based on average ranking score.

**Table 3 healthcare-13-03288-t003:** Areas of additional interest.

	2024	2025
Most important	Medical system information	Community resources
	Support for CHWs	Health topics
	Community resources	Professional development
	Health topics	Medical system information
Least important	Professional development	Support for CHWs

Note: Presented order based on average ranking score.

**Table 4 healthcare-13-03288-t004:** Topic requests.

	2024	2025
Health topics	Pregnancy	Complementary/alternative medicine
	HIV	Autism
	Palliative care	Chronic disease
	Oral health	Smoking cessation
	Eye health	Diabetes
	Autism	Healthy hearts
	Mental health	Asthma
	Nutrition	
	Children’s health	
	LGBTQ+	
	Lead pipes	
	Water safety	
	Sickle cell	
Insurance information	Medicare	Resources for uninsured patients
	Medicaid	Navigating Medicare
	Commercial insurance	
	Applying to insurance	
	Open enrollment	
	SNAP qualifications	
	QMB or SLMB programs	
Community resources	Adult protective services	Resources for unhoused
	Therapy referrals	Transportation for Latino pop.
	UNICEF and WHO	Housing resources
	MTA mobility resources	Partnering with health lefts
	Black Mental Health Alliance	
	Autism Self-Advocacy Network	
	Safe Haven Baltimore	
	Autistic Women and Nonbinary Network	
Professional development	Funding for CBOs	Healthcare system resources
	Expert speakers	Using resources in communities
		Advocacy
		Policy change
		Health literacy
		Effective communication
		Networking support for resources
		Networking support for jobs

Note: HIV = Human Immunodeficiency Virus, LGBTQ+ = Lesbian, Gay, Bisexual, Transgender, Queer or Questioning, and other sexual and gender identities, SNAP = Supplemental Nutrition Assistance Program, QMB = Qualified Medicare Beneficiary, SLMB = Specified Low-Income Medicare Beneficiary, UNICEF = United Nations International Children’s Emergency Fund, WHO = World Health Organization, MTA = Maryland Transit Administration, CBO = Community-Based Organization.

**Table 5 healthcare-13-03288-t005:** Examples of information learned by topic area.

Topic Area	Lunch and Learn Session Topic	One New Piece of Information Learned Today
Health	Autism	“Information about autism and how to be sensitive/culturally competent when navigating neurodivergence.”
		“I learned that Stemming can be calming or due to anxiety.”
	Alzheimer’s disease	“The information on how to get treated for early detection of Dementia.”
		“Risk factors for Alzheimer can be your lifestyle.”
	Breast cancer	“That men can get breast cancer.”
		“Additional imaging required for dense breasts.”
	Lead pipes and water	“That tap water is good for you [in Baltimore].”
		“I learned how our water Federal government works with local governments to make sure water sources and utilities work together to keep us safe.”
Resources	MyChart	“I have learned how to show patients how to navigate in my chart.”
		“Learning how to in depth navigate MyChart and better understanding how to work the program. I did not have much experience previously so it was a nice general overview.”
	Pathfinders for Autism	“More resources for help with patients diagnosed with Autism which I have a huge issue with obtaining information for patients & families.”
		“Resources on where to get help with finding help with autism. Also signs and what to look for.”
Professional development/	CHW burnout	“Strategies to manage burnout.”
CHW Support		“Thinking of a box while taking deep breaths (4 and 4) was interesting and useful.”
	Health literacy	“I learned information in Health literacy and medical messaging.”
		“The real meaning of Health Literacy and Navigation.”

Note. Because feedback was anonymous and aggregate, individual coding was not applicable.

## Data Availability

The data presented in this study are available on request from the corresponding author due to containing identifying information.

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
