# Peer review of "Supporting the Community’s Health Advocates: Initial Insights into the Implementation of a Dual-Purpose Educational and Supportive Group for Community Health Workers"

_healthcare, 2025, doi:10.3390/healthcare13243288_

Round 1
Reviewer 1 Report
Comments and Suggestions for Authors
This is a fantastic practice-based reflection on a bespoke, peer-based mixed virtual/in-person supervision/networking/collaboration/CPD model (Lunch-Learn programme) for a diverse workforce, with many innovative features like ‘holding space’, ‘rolodex’ – including rapid evaluation cycles, goal- and person-centered repeat surveys, and good use of exiting secondary data for formative and summative evaluation in real-time. Impacts of this are well described and evidences good substantiation in both the how (e.g. vicarious loads) and the so-what (esp. up-skilling, overall quality improvement, while lesser so on harder-to-attribute impacts related to care coordination or on service users like ‘bridging’ to be explore in future).
Limitations: consider discussing the prospect of using existing secondary data like staff recruitment/retention/wellbeing data to evidence longer-term impacts.
Landing from Mars: In order to increase relevance to an international audience maybe describe the history of CHWs (Cuba, Brazil door-knocking model of primary care) “Community health workers bring value and deserve to be valued too:” Key considerations in improving CHW career advancement opportunities - PMC; distinction globally: What do we know about community health workers? A systematic review of existing reviews;
maybe also recent policy drives in the UK around assertive outreach NHS England » ‘Saviours’: how community health and wellbeing workers are helping to tackle health inequalities in England
Theory and wider evidence to consider in discussion: mindlines (Gabbay/LeMay); community of practice
Author Response
Reviewer 1 Comments
Comment 1
This is a fantastic practice-based reflection on a bespoke, peer-based mixed virtual/in-person supervision/networking/collaboration/CPD model (Lunch-Learn programme) for a diverse workforce, with many innovative features like ‘holding space’, ‘rolodex’ – including rapid evaluation cycles, goal- and person-centered repeat surveys, and good use of exiting secondary data for formative and summative evaluation in real-time. Impacts of this are well described and evidences good substantiation in both the how (e.g. vicarious loads) and the so-what (esp. up-skilling, overall quality improvement, while lesser so on harder-to-attribute impacts related to care coordination or on service users like ‘bridging’ to be explore in future).
Limitations: consider discussing the prospect of using existing secondary data like staff recruitment/retention/wellbeing data to evidence longer-term impacts.
Response: We thank the reviewer for this encouraging feedback and valuable suggestion. We have now added discussion of potential use of existing secondary workforce data (e.g., retention and well-being metrics) as a strategy for assessing long-term program impacts. This addition appears in Section 4.5 Future Directions/Implications, where we note plans to link Lunch and Learn participation data with workforce retention and well-being indicators in future phases. We have also added to the Limitations (Section 4.4), noting that while initial findings demonstrate engagement and professional growth, future research should explore downstream outcomes such as care coordination effectiveness and client-level impacts (“bridging”). This addition clarifies that such measures were beyond the scope of this proof-of-concept paper but remain a priority for future study.
Comment 2
Landing from Mars: In order to increase relevance to an international audience maybe describe the history of CHWs (Cuba, Brazil door-knocking model of primary care) “Community health workers bring value and deserve to be valued too:” Key considerations in improving CHW career advancement opportunities - PMC; distinction globally: What do we know about community health workers? A systematic review of existing reviews; maybe also recent policy drives in the UK around assertive outreach NHS England » ‘Saviours’:
how community health and wellbeing workers are helping to tackle health inequalities in England.
Response: We appreciate this suggestion and have expanded the Introduction (paragraph 1) to situate CHWs within a global context. We added examples from Latin America and the United Kingdom and added appropriate citations (Early et al., 2016; Junghans et al., 2023).
Comment 3
Theory and wider evidence to consider in discussion: mindlines (Gabbay/Le May); community of practice.
Response: We agree and have incorporated both theoretical frameworks. We added a section to the discussion (Section 4.3 Global Relevance and Theoretical Framework) to help address this. We now explicitly reference mindlines (Gabbay & Le May, 2004; Wieringa & Greenhalgh, 2015) and communities of practice (Noar et al., 2023) to interpret how peer learning and collective knowledge-sharing underpin the Lunch and Learn model. These frameworks strengthen the theoretical grounding of the manuscript and clarify its broader applicability.
Reviewer 2 Report
Comments and Suggestions for Authors
This paper presents a timely and well-intentioned initiative to support community health workers (CHWs) through a structured educational and peer engagement program. The authors clearly demonstrate the program’s relevance and responsiveness to CHW needs, and the integration of participant feedback is commendable. However, the work will benefit from a more critical approach in the writing. For example, the program design is clearly described, however, are not in alignment with current research standards. Also, the discussion is rich in narrative, however, it occasionally lacks analytical depth. Nonetheless, the paper offers a valuable foundation for future research and program development in CHW workforce support.
Here are some key areas to focus on in improving the paper.
Abstract does not meet conventions of the journal, as a result it does not provide appropriate or adequate information requisite for an abstract. Please revise accordingly
Line 46 and 47 (and elsewhere in the document) the use of double dashes (is this meant to be an m-dash?) is unusual and does not meet conventions of the journal
Line 55 – statement needs evidence. Reference needed
Line 85-90 – where is the aim? This needs to be clearly articulated ( I found it in methods). Also line 88 states this is a report, however, it is an article. Has this come from a report? Not clear.
Line 93 – what research method was used? There is no decription… was it an evaluation of the program or something else?
Line 95 – provides background of the program ‘Lunch and Learn’, however methods should be focused on the methods of the evaluation of the program. Suggest that the ‘Lunch and Learn' section be separate after the methods, and the section be focussed on the program itself…
Line 107 – IRB should be the final section of the methods under its own heading. Where is the IRB number and date it was approved? Needs to be transparent as possible. How did people consent? How did you keep individual data confidential?
Line 128 – the survey – this needs to talk about how the survey was developed, tested, including face and content validity etc. How did you know the questions were asking what they were meant to be asking? Where is the participant section, sample size, recruitment method etc. Data collection and data analysis sections seem to be mixed with other sections or absent.
The methods section is not merely a description of what was done; it serves as a demonstration of methodological rigor. By ensuring reliability and repeatability, it enables other researchers to replicate the study under similar conditions and, ideally, arrive at comparable results. This transparency is essential for validating findings and contributing to the broader body of scientific knowledge.
Line 131 – “listserv”… is this a spelling mistake? It is found elsewhere, so this may be a specific term being used, however, in the first instance it needs to be defined, so your audience understands what it is you are referring to.
Line 132 – this is results, and should not be in methods
Line 143 – Zoom data. How was this collected? Needs detail in a data collection section of the methods
Line 163 – who stated this? What codes were developed to link to responses. Why are codes not here? All quotations should be coded or using a pseudonym
Table 3 and 4 – Rankings – how were they ranked/scored. This detail is missing in method and in the tables, what scores did they get to get to the conclusion of the ranking of each item. Not clear?
Line 176 and 183 – are these a heading?
Table 5 is off the page –
All tables are not provided using the template’s requirements and do not mee the journal’s conventions
Table 6 questions need to be linked to participants either through a predetermined code (participate X, 2024) OR a pseudonym
Figure 1 and 2 need to be clearer. Also Table 2 – remove speech bubble – odd to have included in an academic paper…
Discussion – first two paragraphs are not backed up with reputable references. A statement made without evidence is here say or opinion
Discussion is way too long – please make more succinct and backed up with evidence. Here are some key points to focus on
- Clarify the Unique Contribution: Emphasize more clearly how this program differs from or builds upon existing CHW support models. This would help readers understand its innovation and potential for replication.
- Strengthen the Link to Policy: Consider elaborating on how findings could inform policy or funding decisions at the state or national level to support CHW integration and sustainability.
- Address Underrepresented Competencies: The discussion could benefit from a deeper reflection on why certain CHW core competencies (e.g., ethics, communication) were less frequently addressed and how future sessions might fill those gaps.
- Expand on Limitations: Briefly acknowledge limitations of the program or study (e.g., small sample size, self-reported data, limited geographic scope) to enhance transparency and rigor.
- Future Research Directions: Expand on specific research questions or methodologies that could be used in future evaluations (e.g., longitudinal tracking, mixed-methods studies, or comparative analyses with other states).
Other conventions required as part of the article are not present at the end of the document just prior to the reference list. These should be included as part of the of the instructions of the template provided.
Overall, the project is good but could benefit from additional support in the write up. Please do not be disheartened. We need to see good research published, so with some academic help to meet research writing requirements and conventions it will be a great paper!
Author Response
Reviewer 2 Comments
Comment 1
Abstract does not meet conventions of the journal, as a result it does not provide appropriate or adequate information requisite for an abstract. Please revise accordingly.
Response: The abstract has been rewritten in structured format (Background/Objectives, Methods, Results, Conclusions) following Healthcare guidelines.
Comment 2
Line 46 and 47 (and elsewhere in the document) the use of double dashes (is this meant to be an m-dash?) is unusual and does not meet conventions of the journal
Response: All double dashes were removed.
Comment 3
Line 55 – statement needs evidence. Reference needed
Response: The statement was modified, and a supporting citation has been added to substantiate it (reference 9).
Comment 4
Line 85-90 – where is the aim? This needs to be clearly articulated (I found it in methods). Also line 88 states this is a report, however, it is an article. Has this come from a report? Not clear.
Response: The aim is now explicitly stated at the end of the Introduction. Language was adjusted to clarify that this is a manuscript reporting a proof-of-concept project. The word “report” was removed.
Comment 5
Line 93 – what research method was used? There is no decription… was it an evaluation of the program or something else?
Response: We clarified that this is a practice-based implementation description, not a formal evaluation. The Methods section now refers to a developmental assessment approach used to guide real-time program refinement.
Comment 6
Line 95 – provides background of the program ‘Lunch and Learn’, however methods should be focused on the methods of the evaluation of the program. Suggest that the ‘Lunch and Learn' section be separate after the methods, and the section be focused on the program itself…
Response: The 'Program Description' and ‘Developmental Evaluation Approach' are now separate subsections for clarity. (Methods § 2.1 and § 2.2)
Comment 7
Line 107 – IRB should be the final section of the methods under its own heading. Where is the IRB number and date it was approved? Needs to be transparent as possible. How did people consent? How did you keep individual data confidential?
Response: We added a dedicated Ethics and IRB Statement at the end of Methods with the IRB number, consent process, and confidentiality protections.
Comment 8
Line 128 – the survey – this needs to talk about how the survey was developed, tested, including face and content validity etc. How did you know the questions were asking what they were meant to be asking? Where is the participant section, sample size, recruitment method etc. Data collection and data analysis sections seem to be mixed with other sections or absent. The methods section is not merely a description of what was done; it serves as a demonstration of methodological rigor. By ensuring reliability and repeatability, it enables other researchers to replicate the study under similar conditions and, ideally, arrive at comparable results. This transparency is essential for validating findings and contributing to the broader body of scientific knowledge.
Response: We expanded this section to describe the survey’s structure (ranked and open-ended questions), scoring process in Qualtrics, and inclusion criteria. Because this was a formative implementation project rather than a research study, formal psychometric validation and participant demographic collection were not conducted; this rationale is now clearly stated. We also have made the complete surveys available in Supplementary Materials.
Comment 9
Line 131 – “listserv”… is this a spelling mistake? It is found elsewhere, so this may be a specific term being used, however, in the first instance it needs to be defined, so your audience understands what it is you are referring to.
Response: “Listserv” is a standard word in the dictionary. For clarity, we have also defined the word in the manuscript upon first mention.
Comment 10
Line 132 – this is results, and should not be in methods
Response: This content was relocated to the Results section (§3.1).
Comment 11
Line 143 – Zoom data. How was this collected? Needs detail in a data collection section of the methods
Response: Expanded the description under “Engagement Data” (Methods Section §2.3) to explain how Zoom data was collected.
Comment 12
Line 163 – who stated this? What codes were developed to link to responses. Why are codes not here? All quotations should be coded or using a pseudonym
Response: Because open-ended responses were only reviewed descriptively (not formally coded), we added clarification that no coding schema was developed; responses were used to identify broad themes and notable observations (Methods §2.2).
Comment 13
Table 3 and 4 – Rankings – how were they ranked/scored. This detail is missing in method and in the tables, what scores did they get to get to the conclusion of the ranking of each item. Not clear?
Response: A detailed explanation of ranking and scoring in Qualtrics was added to the Methods (§2.2).
Comment 14
Line 176 and 183 – are these headings?
Response: Section headings have been standardized and clearly formatted.
Comment 15
Table 5 off the page; all tables should meet template requirements.
Response: All tables were reformatted to comply with Healthcare template requirements and adjusted for readability. (Tables 1–5)
Comment 16
Table 6 – questions should be linked to participants via code or pseudonym.
Response: Because the feedback was anonymous and aggregated, individual coding was not applicable. A note was added to clarify this. (Table 5 note)
Comment 17
Figure 1 and 2 need to be clearer. Also Table 2 – remove speech bubble – odd to have included in an academic paper…
Response: Figures 1–2 were redrawn with labeled axes, clearer legends, and higher resolution. The speech bubble was removed from Table 2.
Comment 18
Discussion – first two paragraphs are not backed up with reputable references. A statement made without evidence is here say or opinion
Discussion is way too long – please make more succinct and backed up with evidence. Here aresome key points to focus on
Response: The Discussion was reorganized for clarity and suggested key points were expanded on. Each section now includes evidence and citations.
Comment 19
Add back-matter sections required by the journal.
Response: Added all required sections.
Reviewer 3 Report
Comments and Suggestions for Authors
The topic of the article is current and relevant to community and public health.
The methodological description shows some need for improvement to ensure methodological rigor in this type of article.
The following methodological limitations are identified:
Insufficient transparency of instruments—the survey instruments (complete questions, scales) and their validity are not presented. (Sections 2.2 and 3). Lack of detail in data analysis — the analysis appears descriptive; formal methods for qualitative analysis (e.g., thematic analysis approach, coding, triangulation) and quantitative analysis (statistical tests, confidence intervals) are lacking.
Small sample — low number of survey respondents; potential selection bias (email list, volunteers). No description of inclusion/exclusion criteria or response rate relative to the universe of CHWs contacted.
Lack of demographic data — absence of sociodemographic/professional characterization (age, gender, years of experience, type of organization/working conditions) that would allow for interpretation of the study's findings in general.
Suggestions:
Instruments: Attach the complete questionnaires (2024 and 2025) and post-session polls as Appendix/Supplementary Materials. Indicate question type (closed, Likert, open), response options, and validation.
Collection and sampling process: Describe: How participants were recruited (who received the invitation; target universe), response rate (n of invitations vs. n of responses), inclusion/exclusion criteria; Integrate respondents' demographic/professional data (age, gender, years of experience, type of employer).
Quantitative analysis: Include appropriate statistical analyses: descriptive with n and percentages, medians, time trend tests (compare 2024 vs. 2025) with nonparametric/parametric tests as appropriate. Provide confidence intervals and p-values when making claims of significant change.
Qualitative analysis: Describe the analytical approach used for open-ended responses (e.g., Braun & Clarke's thematic analysis, grounded theory, content analysis). Indicate: number of coders, coding process, software used (NVivo/ATLAS.ti/Excel), criteria for saturation, measures of reliability among coders.
Regarding the results, useful indicators (growth of the list, average number of participants, average time spent during the session, frequency of skills covered) are identified as positive points. There are qualitative examples (excerpts from responses) that illustrate perceptions. However, it is suggested to improve the introduction of tables with detailed statistics (e.g., distribution per session, percentage per skill, complete answers to closed questions); to analyze the open-ended responses, as was done with the analytical methodology: how were categories and subcategories extracted, examples of recording units, who coded them, what was the validation process? The graphs referred to (Figure 1 and Figure 2) are not legible and need clear axes and captions.
As for the interpretation and conclusions, there is partial agreement. The conclusions (the program is promising, increases engagement, addresses skills, and provides support) are plausible based on the descriptive data presented. However, due to the lack of robust analysis (and limited sample size), the conclusions should be more cautious regarding the effectiveness and impact on burnout or care outcomes.
With regard to ethics, the article indicates approval by the Johns Hopkins Institutional Review Board and mentions compliance with the Declaration of Helsinki. However, it would be advisable to explicitly mention informed consent (modality) and confidentiality/anonymization measures.
The bibliography is considered adequate and up-to-date, but it is recommended that more recent studies on the evaluation of CHW support interventions.
Author Response
Reviewer 3 Comments
Comment 1
Insufficient transparency of instruments—the survey instruments (complete questions, scales) and their validity are not presented. (Sections 2.2 and 3).
Response: We have added a detailed description of the survey instruments to §2.2 (Methods). The complete 2024 and 2025 annual surveys, as well as the post-session polls, are now included as Supplementary Materials 1-2, specifying question types (ranked-choice, open-ended). Because this project was a formative, practice-based implementation, the instruments were not psychometrically validated; this is now explicitly stated (Methods 2.2).
Comment 2
Lack of detail in data analysis — the analysis appears descriptive; formal methods for qualitative analysis (e.g., thematic analysis approach, coding, triangulation) and quantitative analysis (statistical tests, confidence intervals) are lacking.
Response: Because of small sample size and descriptive design, no formal inferential statistical tests were applied. For open-ended feedback, we added that responses were reviewed descriptively to identify general themes and notable observations; no formal coding or software-assisted thematic analysis was conducted, consistent with the project’s developmental scope (Methods §2.2.).
Comment 3
Small sample — low number of survey respondents; potential selection bias (email list, volunteers).
Response: We acknowledge the small sample size and have added this as a limitation in the Discussion (Limitations §4.4). As the intent was to gather formative feedback from CHWs participating in the Lunch and Learn series rather than to produce generalizable findings, this is appropriate for the project’s proof-of-concept design.
Comment 4
No description of inclusion/exclusion criteria or response rate relative to the universe of CHWs contacted.
Response: Section 2.3 (Methods) was expanded to clarify that invitations were distributed to all members of the CHW Lunch and Learn listserv. Response rates are now reported in Section 3.1 (Survey Completion).
Comment 5
Lack of demographic data — absence of sociodemographic/professional characterization (age, gender, years of experience, type of organization/working conditions) that would allow for interpretation of the study's findings in general.
Response: We appreciate this point. Because the surveys were intended to inform program development rather than research analysis, demographic data were not collected. This limitation is now clearly noted in both the Methods and Limitations sections. Future phases of program evaluation will include demographic and workforce variables to enhance interpretability.
Comment 6
Instruments: Attach the complete questionnaires (2024 and 2025) and post-session polls as Appendix/Supplementary Materials. Indicate question type (closed, Likert, open), response options, and validation.
Response: As suggested, we have added the 2024 and 2025 annual surveys and post-session polls as Supplementary Materials 1-2, including notes on question type (closed, ranked, open).
Comment 7
Collection and sampling process: Describe: How participants were recruited (who received the invitation; target universe), response rate (n of invitations vs. n of responses), inclusion/exclusion criteria; Integrate respondents’ demographic/professional data (age, gender, years of experience, type of employer).
Response: Please see responses to comments 3-5 above.
Comment 8
Quantitative analysis: Include appropriate statistical analyses: descriptive with n and percentages, medians, time trend tests (compare 2024 vs. 2025) with nonparametric/parametric tests as appropriate. Provide confidence intervals and p-values when making claims of significant change.
Response: We clarified that no formal hypotheses driven analyses or quantitative analyses were conducted. No claims of significant change were made.
Comment 9
Qualitative analysis: Describe the analytical approach used for open-ended responses (e.g., Braun & Clarke's thematic analysis, grounded theory, content analysis). Indicate: number of coders, coding process, software used (NVivo/ATLAS.ti/Excel), criteria for saturation, measures of reliability among coders.
Response: The open-ended feedback was reviewed descriptively rather than coded thematically. We have clarified this approach in the Methods. In the discussion, we noted survey responses were analyzed manually and no formal coding software, multiple coders, or reliability measures were used.
Comment 10
The graphs referred to (Figure 1 and Figure 2) are not legible and need clear axes and captions.
Response: All tables were reformatted per journal guidelines.
Comment 11
The graphs referred to (Figure 1 and Figure 2) are not legible and need clear axes and captions.
Response: Figures 1 and 2 were redrawn with clearly labeled axes, improved resolution, and expanded captions to describe the data presented. (Figures 1–2).
Comment 12
As for the interpretation and conclusions, there is partial agreement. The conclusions (the program is promising, increases engagement, addresses skills, and provides support) are plausible based on the descriptive data presented. However, due to the lack of robust analysis (and limited sample size), the conclusions should be more cautious regarding the effectiveness and impact on burnout or care outcomes.
Response: We revised the Conclusions to reflect a more cautious interpretation.
Comment 13
With regard to ethics, the article indicates approval by the Johns Hopkins Institutional Review Board and mentions compliance with the Declaration of Helsinki. However, it would be advisable to explicitly mention informed consent (modality) and confidentiality/anonymization measures.
Response: We added a section to include the Ethics and IRB Statement (Methods §2.4). We also specified that participation in surveys and Zoom sessions was voluntary and that no sensitive information was collected.
Comment 14
The bibliography is considered adequate and up-to-date, but it is recommended that more recent studies on the evaluation of CHW support interventions.
Response: We added an additional recent reference on CHW training and support program evaluation( Reference 27).
Round 2
Reviewer 2 Report
Comments and Suggestions for Authors
Well done on the revisions!!
Happy with the changes.
Author Response
Reviewer 2
Comment: “Well done on the revisions!! Happy with the changes.”
Response:
We greatly appreciate the reviewer’s positive feedback and are delighted that the revisions addressed the concerns raised in the first round of review. Thank you for your supportive and encouraging comments.
Reviewer 3 Report
Comments and Suggestions for Authors
Compared to the previous version, the text has been substantially improved in terms of structure and writing. However, there are still methodological limitations that need to be improved:
Sections 2.1–2.3 describe the context of the program and the tools used well. However, the description is more narrative than systematic, which makes reproducibility difficult. We propose adopting more objective subheadings (e.g., “Participants,” “Data Collection Instruments,” “Analysis Procedures”), in accordance with MDPI guidelines, to better differentiate each component.
Regarding inclusion/exclusion criteria: these refer to the exclusion of participants who spent less than 15 minutes in the session, which requires theoretical justification.
Regarding the description of the samples and responses to the surveys:
The text presents the total number of respondents (n=18 in 2024; n=23 in 2025), but does not provide demographic characteristics, training, or length of service of the CHWs. Even if the study did not collect complete sociodemographic data, it is important to explicitly mention this limitation in item 4.4 (Limitations), detailing how this restricts the generalization of the results.
Regarding instruments and validity, the authors acknowledge that the instruments were not psychometrically validated, but the text does not describe the process of developing the questionnaires (how the questions were created and whether there was a pretest). We suggest adding a short paragraph explaining the origin of the questions (based on literature, previous experiences, or constructed internally) and justifying the use of a descriptive approach.
With regard to data analysis, the methodology mentions only descriptive analyses (frequencies, averages, and automatic rankings in Qualtrics), but does not specify interpretation criteria. It is suggested that the software used (e.g., Excel, Qualtrics exported to SPSS) and how the qualitative results were synthesized (e.g., simple thematic analysis) be indicated.
With regard to Table 1, we suggest grouping the topics by theme (e.g., “Health and diseases,” “System resources and navigation,” “CHW support and well-being”). This would increase readability and reduce visual clutter.
As for Table 2, the side-by-side layout of 2024 and 2025 is clear, but the title “Most important / Least important” could be supplemented with a caption explaining whether the order is based on average ranking (score) or frequency. We suggest adding a footnote specifying the method used to calculate the rankings.
Regarding Table 3, please review the numbering—there are two “Table 3”s. In addition, the presentation is dense and could be summarized in a 3-column structure (Year / Category / Examples of themes). Table 4 is missing.
Review formatting of titles and sections: figure and table captions should follow MDPI style (smaller font, centered, “Figure 1.” / “Table 1.”).
Author Response
Reviewer 3
Comment 1: “Sections 2.1–2.3 describe the context of the program and the tools used well. However, the description is more narrative than systematic, which makes reproducibility difficult. We propose adopting more objective subheadings (e.g., ‘Participants,’ ‘Data Collection Instruments,’ ‘Analysis Procedures’), in accordance with MDPI guidelines, to better differentiate each component.”
Response:
We thank the reviewer for this suggestion. We reviewed the MDPI Healthcare template and author guidelines and noted that they do not specify required subheadings for the Methods section. Because this manuscript describes a practice-based, proof-of-concept implementation rather than a traditional research study, the standard empirical subheadings (e.g., “Participants,” “Sampling,” “Data Collection Instruments”) did not fully align with the structure or aims of the work. For this reason, we have retained the current subheadings, which more accurately reflect the developmental and descriptive nature of the project while still ensuring clarity and transparency for readers.
Comment 2: “Regarding inclusion/exclusion criteria: these refer to the exclusion of participants who spent less than 15 minutes in the session, which requires theoretical justification.”
Response:
We have added clarification in Section 2.3 Engagement Data to explain the rationale for this exclusion. Specifically, individuals who joined for less than 15 minutes (approximately 25% of the session) were excluded to ensure that only meaningful attendance was included in analyses. In our program, the first 15 minutes are generally used to give people an opportunity to come online and for announcements.
Comment 3: “Regarding the description of the samples and responses to the surveys: The text presents the total number of respondents (n=18 in 2024; n=23 in 2025), but does not provide demographic characteristics, training, or length of service of the CHWs. Even if the study did not collect complete sociodemographic data, it is important to explicitly mention this limitation in item 4.4 (Limitations), detailing how this restricts the generalization of the results.”
Response:
We appreciate this point and agree that the absence of demographic data limits generalizability. We already mention this in the limitations section: “As such, we did not collect comprehensive demographic data or retention metrics and cannot draw related conclusions.”
Comment 4: “Regarding instruments and validity, the authors acknowledge that the instruments were not psychometrically validated, but the text does not describe the process of developing the questionnaires (how the questions were created and whether there was a pretest). We suggest adding a short paragraph explaining the origin of the questions (based on literature, previous experiences, or constructed internally) and justifying the use of a descriptive approach.”
Response:
We added that the surveys were developed internally and provided a more detailed explanation of this in Section 2.2 (Developmental Evaluation Approach).
Comment 5: “With regard to data analysis, the methodology mentions only descriptive analyses (frequencies, averages, and automatic rankings in Qualtrics), but does not specify interpretation criteria. It is suggested that the software used (e.g., Excel, Qualtrics exported to SPSS) and how the qualitative results were synthesized (e.g., simple thematic analysis) be indicated.”
Response:
We expanded Section 2.2 (Developmental Evaluation Approach) and Section 2.3 (Engagement Data) to clarify that quantitative analyses were descriptive and conducted in Excel using data exported from Qualtrics. For open-ended items, responses were reviewed manually to identify broad themes and notable insights, following a simplified thematic review approach without formal coding software. These additions specify analytic procedures and interpretation criteria.
Comment 6: “With regard to Table 1, we suggest grouping the topics by theme (e.g., ‘Health and diseases,’ ‘System resources and navigation,’ ‘CHW support and well-being’). This would increase readability and reduce visual clutter.”
Response:
We have revised Table 1 to group Lunch and Learn topics thematically under four categories: (1) Health and Disease Topics, (2) System and Resource Navigation, (3) CHW Support and Professional Development, and (4) Community and Organizational Resources. This restructuring improves clarity and readability.
Comment 7: “As for Table 2, the side-by-side layout of 2024 and 2025 is clear, but the title ‘Most important / Least important’ could be supplemented with a caption explaining whether the order is based on average ranking (score) or frequency. We suggest adding a footnote specifying the method used to calculate the rankings.”
Response:
We already clarify in Section 2.2. that “Rankings were scored automatically using Qualtrics software, which assigns a numerical value to each position and aggregates results across respondents to determine the average ranking for each option.” We added that “Lower mean values indicate more importance” for clarity. In addition, have revised Table 2 and 3 to include a caption clarifying the presentation order of the data. (Table 2 and 3 captions)
Comment 8: “Regarding Table 3, please review the numbering—there are two ‘Table 3’s. In addition, the presentation is dense and could be summarized in a 3-column structure (Year / Category / Examples of themes). Table 4 is missing.”
Response:
Thank you for catching this oversight. We have corrected the table numbering and titles throughout the manuscript. While we understand and appreciate the rationale for the suggested reformatting, we believe that maintaining the current structure offers the clearest and most comprehensive presentation of the data.
Comment 9: “Review formatting of titles and sections: figure and table captions should follow MDPI style (smaller font, centered, ‘Figure 1.’ / ‘Table 1.’).”
Response:
We have reviewed all figure and table captions and reformatted them to our best ability according to MDPI’s author guidelines—centered, with appropriate capitalization and numbering. (Tables 1–5; Figures 1–2)
Summary
We thank Reviewer 3 for the thoughtful and detailed feedback. In response, we clarified the Methods section, added justification for inclusion and exclusion criteria, expanded descriptions of survey development and data analysis, and revised tables and captions for clarity. We also noted the limitation related to missing demographic data. These revisions improve the manuscript’s clarity, rigor, and transparency while maintaining its descriptive focus.